# Numerical Investigation on High-Performance Cu-Based Surface Plasmon Resonance Sensor for Biosensing Application

**DOI:** 10.3390/s23177495

**Published:** 2023-08-29

**Authors:** M. Muthumanikkam, Alagu Vibisha, Michael Cecil Lordwin Prabhakar, Ponnan Suresh, Karupiya Balasundaram Rajesh, Zbigniew Jaroszewicz, Rajan Jha

**Affiliations:** 1Department of ECE, Vel Tech Rangarajan Dr Sagunthala R & D Institute of Science and Technology, Chennai 600025, Tamil Nadu, India; muthuroots@gmail.com (M.M.); drlordwin@veltech.edu.in (M.C.L.P.); drpsuresh@veltech.edu.in (P.S.); 2Department of Physics, Chikkanna Government Arts College, Tiruppur 641602, Tamil Nadu, India; alaguvibisha@gmail.com; 3National Institute of Telecommunications, ul. Szachowa 1, 04-894 Warsaw, Poland; 4Nanophotonics and Plasmonic Laboratory, School of Basic Sciences, Indian Institute of Technology, Bhubaneswar 752050, Odisha, India; rjha@iitbbs.ac.in

**Keywords:** surface plasmon resonance, biosensor, sensitivity, titanium oxide, copper, barium titanate

## Abstract

This numerical research presents a simple hybrid structure comprised of TiO_2_-Cu-BaTiO_3_ for a modified Kretschmann configuration that exhibits high sensitivity and high resolution for biosensing applications through an angular interrogation method. Recently, copper (Cu) emerged as an exceptional choice as a plasmonic metal for developing surface plasmon sensors (SPR) with high resolution as it yields finer, thinner SPR curves than Ag and Au. As copper is prone to oxidation, especially in ambient conditions, the proposed structure involves the utilization of barium titanate (BaTiO_3_) film as a protection layer that not only preserves Cu film from oxidizing but enhances the performance of the sensor to a great extent. Numerical results also show that the utilization of a thin adhesive layer of titanium dioxide (TiO_2_) between the prism base and Cu film not only induces strong interaction between them but also enhances the performance of the sensor. Such a configuration, upon suitable optimization of the thickness of each layer, is found to enhance sensitivity as high as 552°/RIU with a figure of merit (FOM) of 136.97 RIU^−1^. This suggested biosensor design with enhanced sensitivity is expected to enable long-term detection with greater accuracy and sensitivity even when using Cu as a plasmonic metal.

## 1. Introduction

The SPR sensor is found to be prominent among the various current sensing approaches owing to its reliability, rapid analysis, high sensitivity, accurate detection, and label-free detection method for chemical and biological analytes [1,2,3]. This makes it possible to use the SPR sensor for various purposes, including environmental monitoring, pressure sensing [4], temperature sensing [5], medical diagnosis [6], DNA hybridization detection, discovering drugs [7], spotting molecules, glucose monitoring [8], formalin detected in the food preservatives [9], etc. The SPR method has the virtue of being able to identify even minute variations in the refractive index (RI) of the sensing medium [10]. The attenuated total reflection (ATR) phenomenon is supported by the angular interrogation method, and SPR sensors frequently prefer the angular method because of its simplicity and high resolution [11]. In general, plasmonic materials, including silver (Ag) and gold (Au), have been utilized in SPR sensors. As gold is invulnerable to degradation and corrosion, it is recommended as a superior plasmonic material for the SPR sensor. The drawbacks of Au are that it has low biomolecule adhesion capabilities and provides wider reflectance curves [12,13]. Ag is considered a potential substitute metal since it is less expensive and produces a finer SPR spectrum than Au, but it oxidizes easily [9,14]. A lot of recent research favored copper as a plasmonic metal as it benefits higher electrical conductivity, it has the ability to generate narrow SPR peaks, and it is a cost-effective approach when compared to gold and silver [15,16,17]. The Cu inter-band transition exhibits strong optical absorption properties and is quite similar to Au [18]. Due to its unique aspects, Cu is an ideal choice to use in SPR sensors. Nevertheless, because Cu is an easily and quickly oxidizing metal, using it directly as an individual metal layer will result in dense, fragile oxide layers [19]. Recently, several approaches, including bimetal (such as Cu-Ni) [20], coating metal oxide protection layers (like SiO_2_, Fe_2_O_3_, BaTiO_3_, ZnO, MoO_3_) [16,19,21,22,23], or 2D materials [24] are suggested to overcome the oxidation of copper. It is noted that well-optimized suitable protection layers not only inhibit oxidation but greatly improve the performance of the SPR sensors.

Recently, Mumtaz et al. proposed a fiber-based SPR sensor in which magnetron sputtered is used to coat iron oxide (Fe_2_O_3_) on Ag to protect the Ag layer from oxidation. This inclusion of iron oxide increases the effective absorption coefficient, which in turn improves the SPR sensor’s sensitivity [25]. Additionally, Fe_2_O_3_ is a great prospect for future endeavors in SPR sensor developments since they are environmentally friendly, abundant, air-stable, highly corrosion-resistant, non-toxic, and affordable [26]. Zhang et al. designed a biomolecule detection SPR sensor using Fe_2_O_3_, which enhances the electric field and evanescent field depth at the Fe_2_O_3_/sensing medium interface because Fe_2_O_3_ has a high refractive index value, resulting in highly sensitive performance [27]. Not long ago, Ahmed and Wang et al. reported that the SPR biosensor using Fe_2_O_3_ layer over Au and Cu thin film provided the best sensing performance due to its good adsorbent property and its low extinction coefficient [19,28]. Lately, Augustine et al. reported the superior biocompatibility of molybdenum trioxide (MoO_3_) suitable for the detection of breast cancer [29]. Zakaria et al. developed an SPR sensor with MoO_3_ coated on top of Ag and Cu metal, which shows an increase in sensitivity as well as provides protection against oxidation of plasmonic metals [30]. Pandey et al. suggested an approach to increasing the sensitivity of an SPR sensor by sandwiching MoO_3_ between Ag and MXene [31]. The solid phase deposition of MoO_3_ film on Cu substrate is also reported [32]. Recently, the biocompatibility of the BaTiO_3_ made it a suitable candidature for SPR sensors. Apart from being inexpensive, it possesses a wide range of preparation methods accessible for the nanoparticles’ in-house synthesis [33,34,35]. Ihlefeld et al. presented the synthesis and properties of BaTiO_3_ solid solution thin films deposited via a chemical solution approach on Cu substrates [36]. Recently, several SPR sensor structures utilizing BaTiO_3_ for improved performances have been reported both theoretically and experimentally [37,38,39,40,41,42].

Recent studies show that the use of an adhesive layer on the prism solves the issue of deteriorating sensitivity and improves plasmonic activity to collect incident light effectively [43]. For adhesive layers, recent SPR biosensor research focused on using high-RI metal oxide layers such as TiO_2_ and ZnO [44]. Recently, several studies reported an improvement in the performance of the SPR sensor that utilized TiO_2_ as an adhesive layer at the prism/metal interface [45,46,47,48,49,50,51].

In this numerical work, a modified Kretschmann configuration is proposed in which the advantages of Cu as a plasmonic metal and the benefits of TiO_2_ as an adhesive layer are utilized. Here, the superiority of three different oxide layers (BaTiO_3_, Fe_2_O_3_, and MoO_3_) on improving the sensor performance is analyzed in detail on the basis of numerical results. The layer thickness of each film coating is well optimized with the aim of achieving enhanced sensitivity, reduced full width at half maximum (FWHM) of the SPR spectrum, and achieving minimum reflectance to ensure high sensitivity as well as high FOM.

## 2. Theoretical Model

### 2.1. Structure Description

A schematic representation of the modified Kretschmann configuration utilizing a five-layer configuration (BK7 prism, TiO_2_, Cu, oxide layer, and sensing layer) for biosensing application is shown in Figure 1. The RI of the first layer (BK7 prism) is 1.5151 [52]. The RI of the second layer (TiO_2_) is 2.5837 [48]. The third layer (Cu) is coated on the TiO_2_ layer.

The dielectric constant of metal (Cu) is obtained using the Drude model and is given by Equation (1)
(1)$$\varepsilon_{m}\left(\lambda \right)=\varepsilon_{mr}+\varepsilon_{mi}=1-\frac{\lambda^{2}\lambda_{c}}{\lambda_{p}^{2}\left(\lambda_{c}+i\lambda \right)},$$
where *ε_mr_* and *ε_mi_* represent the metal layer dielectric constant real part and imaginary part, respectively. For Cu: plasma wavelength (*λ_p_*) = 0.13617 × 10^−6^ m and collision wavelength (*λ_c_)* = 40.852 × 10^−6^ m [16]. The fourth layer (oxide layer) used to inhibit the oxidation of Cu metal and its RI is given in Table 1.

The fifth layer is the sensing zone, whose RI is assumed to change in the range of *n_s_* = 1.33 to *n_s_* = 1.33 + *δn*, where *δn* denotes the change in RI of the sensing medium due to the adsorption of biomolecules. The range of change in refractive index (*δn*) for biomolecular adsorption is typically on the order of 0.005. This means that even tiny amounts of biomolecules binding to the sensor’s surface can lead to detectable shifts in the SPR signal. The specific range of change in RI depends on factors like the size and mass of the biomolecules, the density of the immobilized ligands, and the interactions between the molecules themselves. Here, we assumed the adsorption of biomolecules occurred on the surface of the metal oxide layer, and the refractive index of the sensing medium changed from 1.33 to 1.335 [54].

### 2.2. Reflectance

The reflectance of incident light (p-polarized) is calculated for this multi-layer structure using the transfer matrix method [55]. The tangential fields at the first boundary *Z* = *Z*_1_ = 0 and the tangential fields at the last boundary *Z* = *Z_N_*_−1_ are related by
(2)$$\left[\begin{array}{c} U_{1}\\ V_{1} \end{array}\right]=M\left[\begin{array}{c} U_{N-1}\\ V_{N-1} \end{array}\right]$$
where *U*_1_ and *V*_1_ are the tangential components of electric and magnetic fields, respectively, at the first layer boundary. *U_N_*_−1_ and *V_N_*_−1_ are the corresponding fields at the *N*th layer boundary. *M* refers to the characteristic matrix of the *N*-layer model and is given by
(3)$$M=\overset{N-1}{\underset{k=2}{\Pi }}M_{k}=\left[\begin{array}{cc} M_{11} & M_{12}\\ M_{21} & M_{22} \end{array}\right]$$
with
(4)$$M_{k}=\left[\begin{array}{cc} \cos\beta_{k} & \frac{-i\sin\beta_{k}}{q_{k}}\\ -iq_{k}\sin\beta_{k} & \cos\beta_{k} \end{array}\right]$$

The phase factor (*q_k_*) and optical admittance (*β_k_*) of the *k*th layer are expressed by
$$q_{k}={\left(\frac{\mu_{k}}{\varepsilon_{k}}\right)}^{\frac{1}{2}}cos\theta_{k}=\frac{{\left(\varepsilon_{k}-n_{1}^{2}\sin^{2}\theta_{1}\right)}^{\frac{1}{2}}}{\varepsilon_{k}}\beta_{k}=\frac{2\pi }{\lambda }n_{k}cos\theta_{k}\left(z_{k}-z_{k-1}\right)=\frac{2\pi d_{k}}{\lambda }{\left(\varepsilon_{k}-n_{1}^{2}\sin^{2}\theta_{1}\right)}^{\frac{1}{2}}$$

Here, *θ*_1_, *n*_1_, and *λ* denote the incident angle, RI of the prism, and wavelength of the incident light (633 nm), whereas *μ_k_, ε_k_*, and *d_k_* represent the permeability, dielectric constant, and thickness of the kth layer, respectively.

Reflectance (*R_p_)* and reflection coefficient (*r_p_*) of incident light are given as
(5)$$R_{p}={\left|r_{p}\right|}^{2}$$
(6)$$r_{p}=\frac{\left(M_{11}+M_{12}q_{N}\right)q_{1}-\left(M_{21}+M_{22}q_{N}\right)}{\left(M_{11}+M_{12}q_{N}\right)q_{1}+\left(M_{21}+M_{22}q_{N}\right)}$$

### 2.3. Performance Parameters

#### 2.3.1. Sensitivity (*S_n_)*

Sensitivity is defined as the ratio between the resonance angle switching (*δθ*) and the sensing layer’s *RI* variation (*δn_s_*). It is given as below [13]:(7)$$S_{n}=\frac{\delta \theta }{\delta n_{s}}\left(deg/RIU\right)$$

Even if the *RI* of the sensing layer is just slightly changed, the best SPR sensor will offer the largest changes in resonance angle.

#### 2.3.2. Detection Accuracy (DA) or Signal-to-Noise Ratio (SNR)

The detection accuracy is inversely related to the FWHM (curve width at 50% reflectance) of the reflectance curve [13].
(8)$$DA=\frac{1}{FWHM}({deg}^{-1})$$

The exact spot of the resonance angle can be identified if the reflectance spectrum is narrow (i.e., smaller *FWHM*), which leads to improved detection accuracy.

#### 2.3.3. Figure of Merit (FOM) or Quality Factor (Q)

A figure of merit is an important criterion used to demonstrate sensitivity and *FWHM* impact on the sensor’s performance [43].
(9)$$FOM=\frac{S_{n}}{FWHM}\left(RIU^{-1}\right)$$

#### 2.3.4. Electric Field Intensity Enhancement Factor (EFIEF)

The extent to which the field has been efficiently focused along the BaTiO_3_-analyte interface is shown by the electric field intensity enhancement factor [56,57]. For p-polarized light, the EFIEF is the ratio of the square of the electric field (*E*) or magnetic field (*H*) at the oxide layer/analyte interface to the square of the field E or H at the prism/TiO_2_ interface. EFIEF is expressed by the
(10)$${\left|\frac{E\left(\frac{N}{N-1}\right)}{E\left(\frac{1}{2}\right)}\right|}^{2}=\frac{\varepsilon_{1}}{\varepsilon_{N}}{\left|\frac{H\left(\frac{N}{N-1}\right)}{H\left(\frac{1}{2}\right)}\right|}^{2}=\frac{\varepsilon_{1}}{\varepsilon_{N}}{\left|t\right|}^{2}$$
where *t* is indicated as the transmission coefficient and *ε*_1_ and *ε_N_* are denoted as the dielectric constants of the first layer and the *N*th layer, respectively.

## 3. Results and Discussion

The Fresnel formula and the transfer matrix method are implemented to properly assess all functional parameters. A proper covering layer strategy is required to avoid a Cu-based SPR sensor showing degraded performance due to the oxidation effect of Cu. Here, we suggested to use three different oxide coatings, namely BaTiO_3_, Fe_2_O_3_, and MoO_3_, over Cu to inhibit its oxidation effect [30,38,40], and the proposed model also includes a thin layer of TiO_2_ as an adhesive that strongly binds Cu on the prism base [45,47,48,49,50]. The thickness of every layer is well optimized, which leads to achieve high sensitivity, the lowest minimum reflectance (R_min_), and the thinnest FWHM of the resonance curve [58]. Here, we meticulously examined the role that each layer plays in the proposed configuration.

Initially, we investigated the effect of BaTiO_3_ on copper and found out its optimized thickness to achieve the best sensing performance. Figure 2a shows the change in sensing parameters (sensitivity, R_min_, and FWHM) corresponding to the change in the Cu layer thickness in the range of 20 nm to 60 nm for 5 nm BaTiO_3_ protection coating on the copper layer. It is observed that the FWHM and R_min_ values of the SPR dip decrease as the Cu layer thickness increases, whereas the sensitivity is observed to increase from 120°/RIU to 132°/RIU. It is also noted that the sensor performance is better for the 55 nm thickness of the copper layer as the resonance curve R_min_ value is closer to zero (0.013), with sensitivity around 132°/RIU, and the FWHM as small as 0.72°. Further examining the same configuration using a 10 nm thickness of BaTiO_3_ cover over the Cu layer, 55 nm thickness of Cu again shows maximum sensitivity (178°/RIU) with R_min_ as 0.009 at 1.33RI and 0.0081 at 1.335RI, and the FWHM value as 1.23°. It is also observed that further increasing the thickness of BaTiO_3_ to 15 nm, the 45 nm Cu layer shows an R_min_ value close to zero and exhibits high sensitivity of about 486°/RIU with the FWHM of the resonance curve increased to 4.27°. From the above three cases, it is observed that 15 nm BaTiO_3_ coated on a 45 nm Cu layer provides excellent sensing capability and hence it is considered for further optimization.

In the next phase of optimization, the impact of utilizing TiO_2_ adhesive film between the BK7 prism and the Cu layer is analyzed. Figure 3 shows the same as Figure 2 but for sandwiching a 5 nm thickness of TiO_2_ as an adhesive layer between the prism and Cu film. Here, it is observed that the maximum sensitivity achieved for 5 nm and 10 nm of BatiO_3_ cover layers are 132°/RIU and 180°/RIU, corresponding to the thickness of the Cu layer as 55 nm for both cases. Though the sensitivity remains almost the same as in the previous cases without the TiO_2_ adhesive layer (Figure 2a,b), the FOM for these cases are improved to 191.3 RIU^−1^ and 156.5 RIU^−1^ due to a reduction in the FWHM of the resonance spectrum, as shown in Table 2. Figure 3c shows that upon utilizing a 15 nm thickness of BaTiO_3_, the maximum sensitivity around 552°/RIU with FOM around 136.97 RIU^−1^ is achieved for a 45 nm thickness of Cu. This is because the TiO_2_ layer induces a significant SPR effect when joined with metal as it assists in enhancing surface plasmons (SPs) at the metal/prism interface and hence improves sensitivity and reduces SP damping (FWHM) [59].

In the following phase, we examined the sensor performance for Fe_2_O_3_ protection layer situated over Cu layer. From Figure 4, we found that R_min_ close to zero are obtained at 55 nm, 50 nm, and 45 nm thickness of Cu corresponding to 5 nm, 10 nm, and 11 nm Fe_2_O_3_ layer.

The calculated sensitivities are 142°/RIU, 282°/RIU, 406°/RIU with FOM as 147.9 RIU^−1^, 85.97 RIU^−1^, and 78.07 RIU^−1^, respectively. Moreover, we also analyzed the effect of the MoO_3_ cover layer on Cu film.

Figure 5 shows that for the MoO_3_ layer with thicknesses 5 nm, 10 nm, and 27 nm, the R_min_ values reach values close to zero corresponding to thicknesses of Cu film as 55 nm, 50 nm, and 45 nm, respectively. For these cases, the sensitivity obtained is 118°/RIU, 130°/RIU, and 386°/RIU with the corresponding FOM calculated as 203.4 RIU^−1^, 173.3 RIU^−1^, and 104.89 RIU^−1^, respectively. In the above three cases, the BaTiO_3_ layer achieves maximum sensitivity as it possesses a large real part of the dielectric constant with no imagery part. So, it is the most suitable covering layer when compared to other oxide layers for the proposed sensor. Thus, we optimize that the 45 nm Cu sandwiched between 5 nm TiO_2_ and 15 nm BaTiO_3_ outer cover is a better configuration that achieves sensitivity and FOM as high as 552°/RIU and 136.97 RIU^−1^, respectively, and the effects of each oxide layer (BaTiO_3_, Fe_2_O_3_, and MoO_3_) is also compared in Table 2.

The thickness of the oxide layer plays a critical role in the suggested sensor; thus, we carried out an extensive study to ensure the best possible value, as illustrated in Figure 6. It is noticed that the reflectance spectrum moves to a greater incidence angle as the thickness of the outer layer (BaTiO_3_, MoO_3_, and Fe_2_O_3_) increases. It is also noted that for 15 nm of BaTiO_3_, 27 nm of MoO_3_, and 11 nm of Fe_2_O_3_, the R_min_ obtained is almost zero, and such condition is much favored for maximum conversion of incident light energy into surface plasmons. Further increasing of thickness above the previously prescribed values for all three cover layers R_min_ values increases because the rate of light utilization reduces as the oxide layer thickness increases.

The performance of the SPR biosensor is also determined by field distribution at the interface of the metal/dielectric interface. The interaction between the evanescent field and the biomolecule in the sensing medium is crucial. There is more biomolecular interaction when the field dispersion is improved [56,57]. Figure 7 shows that the EFIEF decreases when the sensing medium RI changes from n_s_ = 1.33 to n_s_ = 1.335. This is because biomolecules strongly bind to the detection surface of the biosensor. The electric field distributions of the optimized configuration of the structure TiO_2_-Cu-BaTiO_3_ is shown in Figure 8. It is noted that the electric field intensity at the interface of Cu-BaTiO_3_ is increasing and reaches its peak at the interface of BaTiO_3_ and the sensing medium. In this proposed structure, the numerical value of the probing field is much more intense in the sensing medium, which leads to a stronger excitation of SP waves, resulting in enhanced sensitivity. Comparing this proposed structure to prior published similar sensors structures, the sensing output is much higher and is compared with others in Table 3.

## 4. Conclusions

This numerical work demonstrates a highly sensitive SPR biosensor with a hybrid configuration made of layers of TiO_2_, Cu, and BaTiO_3_/Fe_2_O_3_/MoO_3_. The thickness of the suggested layers (TiO_2_, Cu, and BaTiO_3_/Fe_2_O_3_/MoO_3_) is carefully tuned to achieve distinctly higher sensitivity as well as FOM. Here, the utilization of the adhesion layer of TiO_2_ enhances light trapping capability, which, in turn, also enhances the sensitivity of the suggested sensor. This proposed structure, when using BaTiO_3_ as a covering layer, attains high sensitivity (552°/RIU) as well as high FOM (136.97 RIU^−1^) when compared to Fe_2_O_3_ (406°/RIU and 78.07 RIU^−1^) and MoO_3_ (386°/RIU and 104.89 RIU^−1^) for the optimized thickness of 45 nm Cu sandwiched between 5 nm TiO_2_ and 15 nm BaTiO_3_ outer cover. The proposed structure is expected to enable long-term detection with greater accuracy and sensitivity even when using Cu as a plasmonic metal. This study offers a novel possibility for the development of a more accurate and highly sensitive biosensor for biological sensing uses.

## Figures and Tables

**Figure 1 sensors-23-07495-f001:**
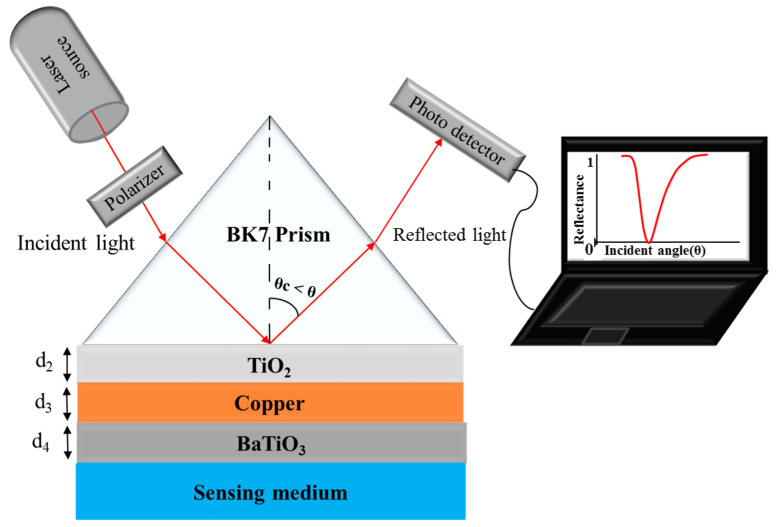
Diagrammatic representation for the hybrid structure of BK7-TiO_2_-copper-BaTiO_3_-based SPR biosensor.

**Figure 2 sensors-23-07495-f002:**
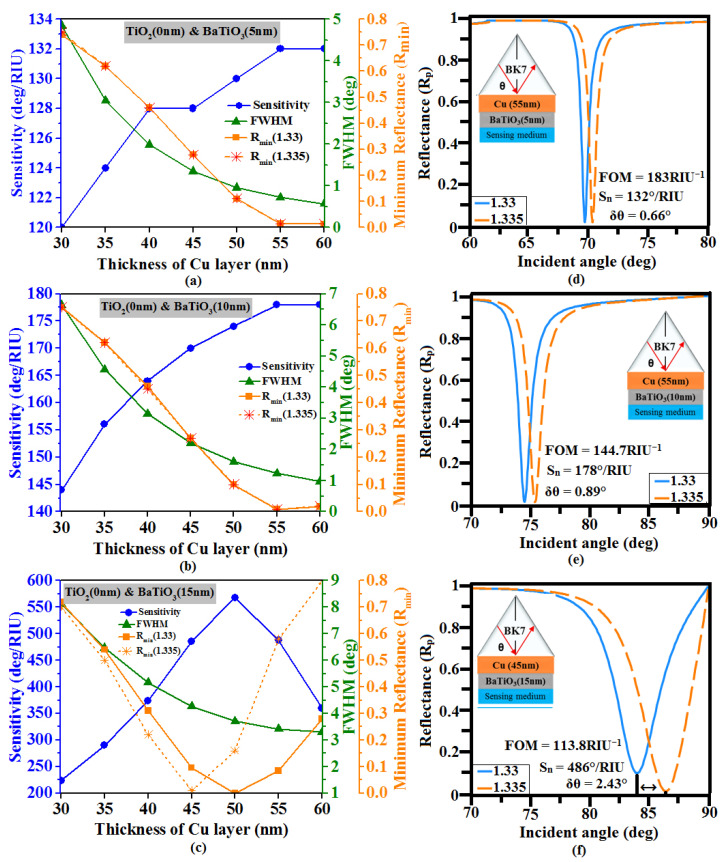
Variation of the sensitivity, minimum reflectance (R_min_), and FWHM vs. thickness of Cu layer (30 nm to 60 nm) for the thickness of BaTiO_3_ layers (**a**) 5 nm, (**b**) 10 nm, and (**c**) 15 nm. Optimized structure SPR curve: (**d**) Cu (55 nm)-BaTiO_3_ (5 nm), (**e**) Cu (55 nm)-BaTiO_3_ (10 nm), and (**f**) Cu (45 nm)-BaTiO_3_ (15 nm).

**Figure 3 sensors-23-07495-f003:**
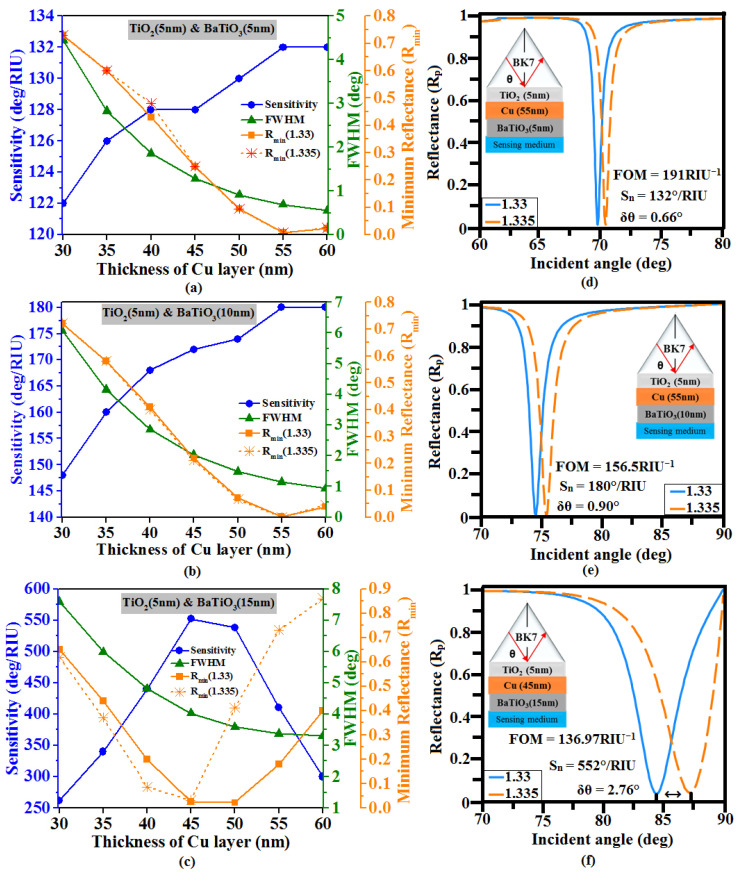
Variation of the sensitivity, minimum reflectance (R_min_), and FWHM vs. thickness of Cu layer (30 nm to 60 nm) for the thickness of BaTiO_3_ layers (**a**) 5 nm, (**b**) 10 nm, and (**c**) 15 nm. Optimized structure SPR curve: (**d**) TiO_2_ (5 nm)-Cu (55 nm)-BaTiO_3_ (5 nm), (**e**) TiO_2_ (5 nm)-Cu (55 nm)-BaTiO_3_ (10 nm), and (**f**) TiO_2_ (5 nm)-Cu (45 nm)-BaTiO_3_ (15 nm).

**Figure 4 sensors-23-07495-f004:**
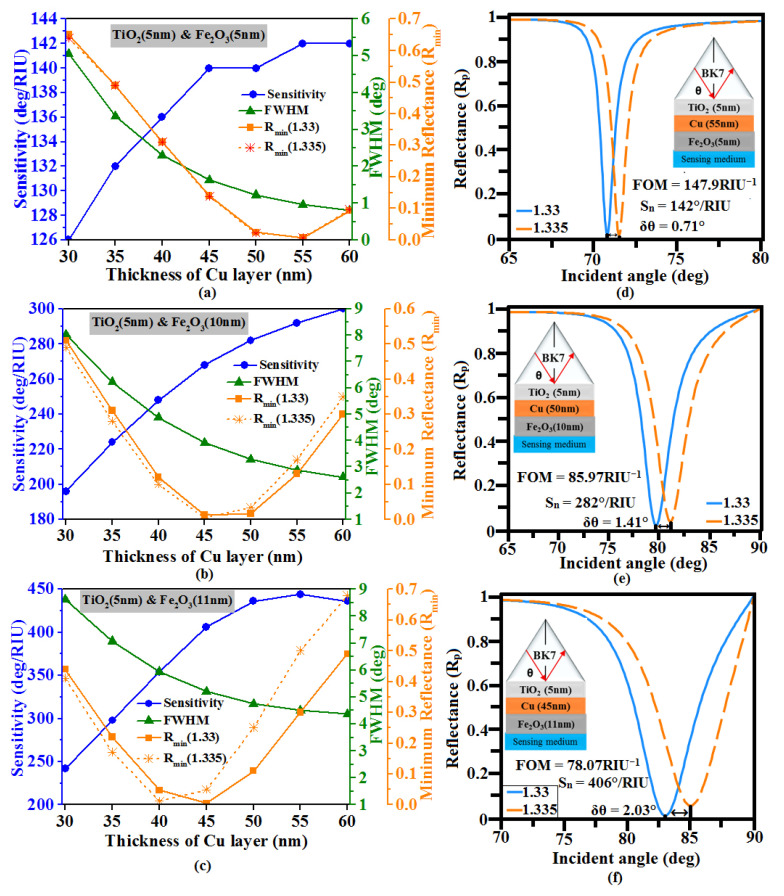
Variation of the sensitivity, minimum reflectance (R_min_), and FWHM vs. thickness of Cu layer (30 nm to 60 nm) for the thickness of Fe_2_O_3_ layers (**a**) 5 nm, (**b**) 10 nm, and (**c**) 11 nm. Optimized structure SPR curve: (**d**) TiO_2_ (5 nm)-Cu (55 nm)-Fe_2_O_3_ (5 nm), (**e**) TiO_2_ (5 nm)-Cu (50 nm)-Fe_2_O_3_ (10 nm), and (**f**) TiO_2_ (5 nm)-Cu (45 nm)-Fe_2_O_3_ (11 nm).

**Figure 5 sensors-23-07495-f005:**
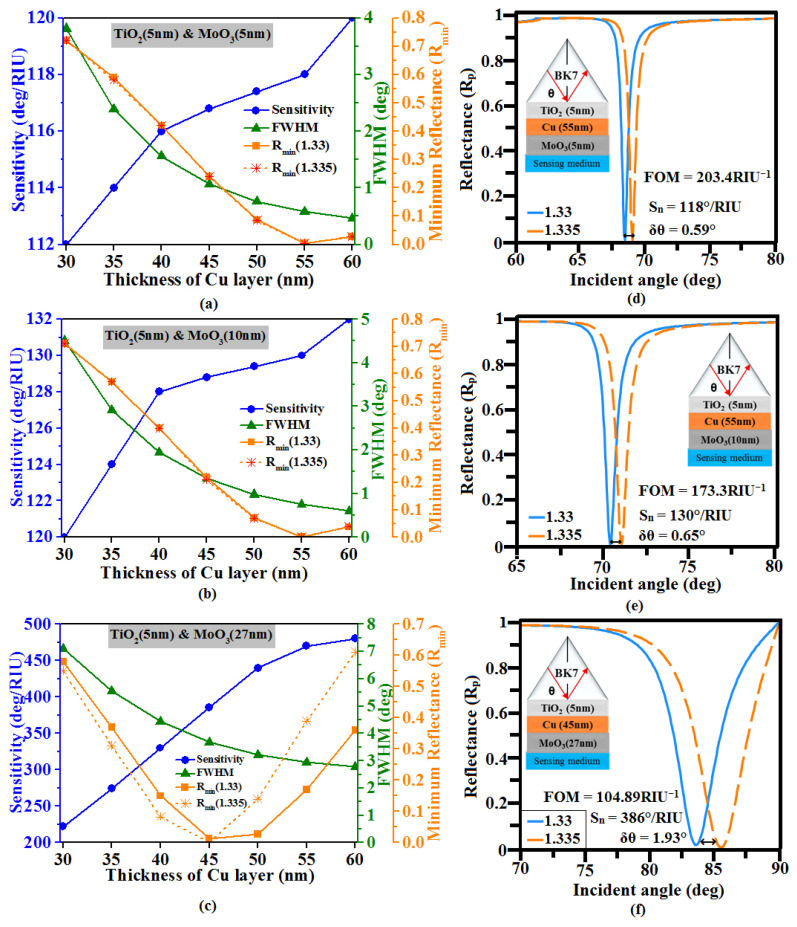
Variation of the sensitivity, minimum reflectance (R_min_), and FWHM vs. thickness of Cu layer (30 nm to 60 nm) for the thickness of MoO_3_ layers (**a**) 5 nm, (**b**) 10 nm, and (**c**) 27 nm. Optimized structure SPR curve: (**d**) TiO_2_ (5 nm)-Cu (55 nm)-MoO_3_ (5 nm), (**e**) TiO_2_ (5 nm)-Cu (55 nm)-MoO_3_ (10 nm), and (**f**) TiO_2_ (5 nm)-Cu (45 nm)-MoO_3_ (27 nm).

**Figure 6 sensors-23-07495-f006:**
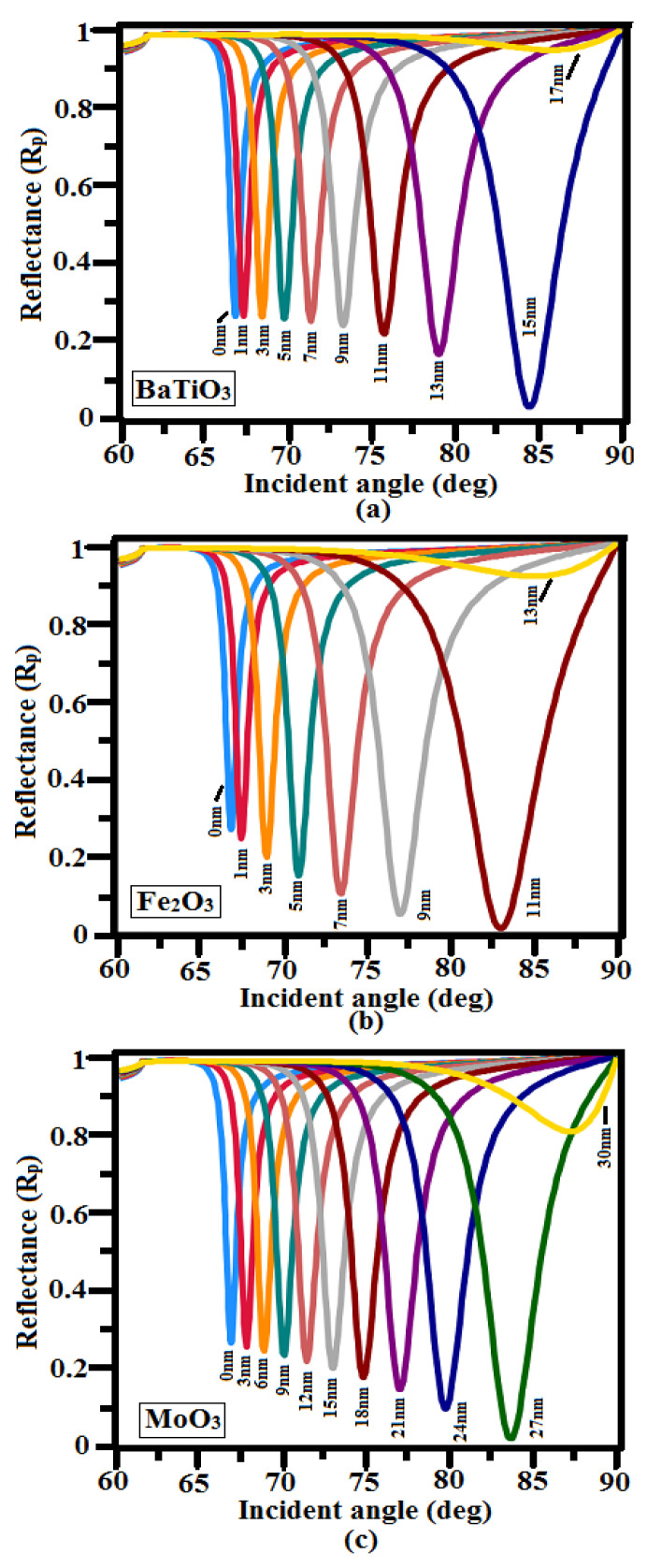
Reflectance vs. the incident angle for the different thicknesses of oxide layers: (**a**) BaTiO_3_, (**b**) Fe_2_O_3_, and (**c**) MoO_3_ with TiO_2_ = 5 nm and Cu = 45 nm at *n*_s_ = 1.33.

**Figure 7 sensors-23-07495-f007:**
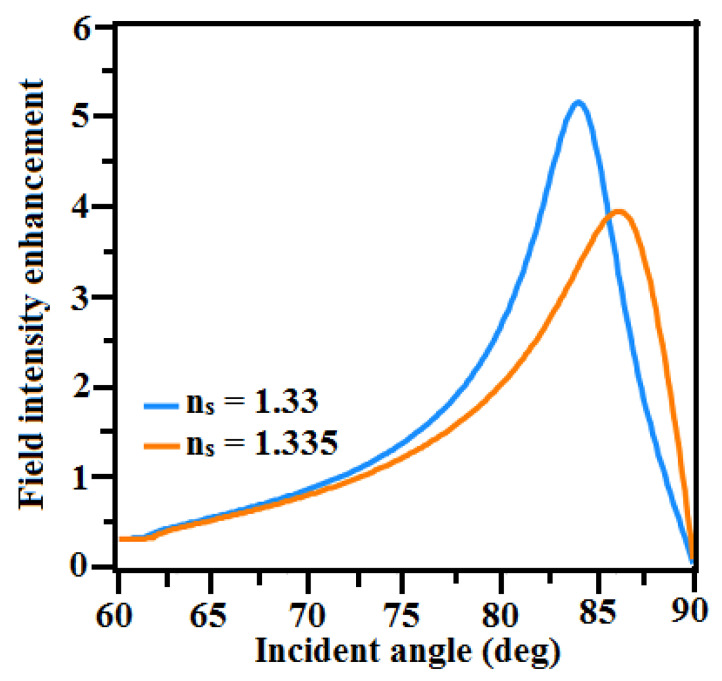
Incidence angle vs. electric field intensity enhancement factor at n_s_ = 1.33 and n_s_ = 1.335 for the proposed TiO_2_-Cu-BaTiO_3_ optimized configuration.

**Figure 8 sensors-23-07495-f008:**
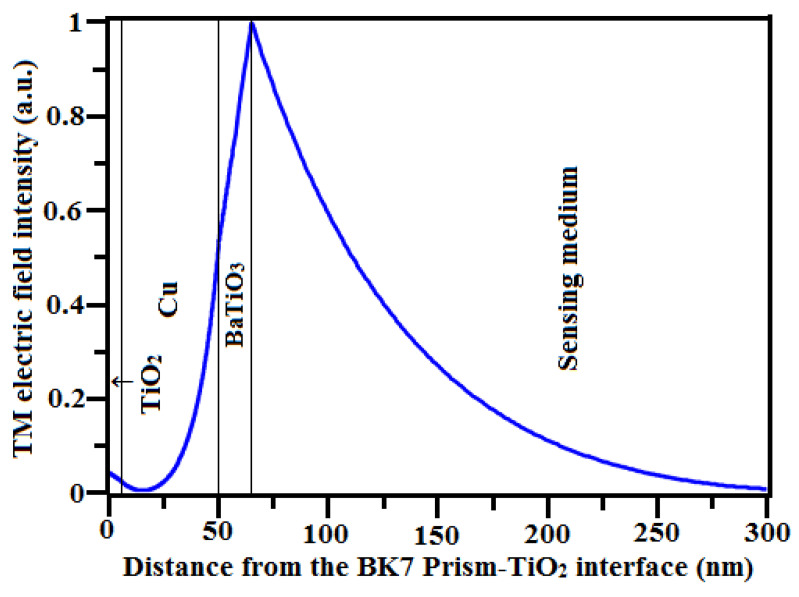
The electric field intensity distributions of the optimized TiO_2_-Cu-BaTiO_3_-based structure for n_s_ = 1.33.

**Table 1 sensors-23-07495-t001:** Refractive index of the oxide layers at *λ* = 633 nm.

Oxide Layer	Refractive Index	Ref.
BaTiO_3_	2.4043	[53]
Fe_2_O_3_	2.918 + 0.029i	[28]
MoO_3_	1.8233 + 0.00204i	[31]

**Table 2 sensors-23-07495-t002:** The oxide layers attain the best results (R_min_, FWHM, sensitivity, and FOM) at the optimized thickness of Cu layer.

Thickness of Oxide Layers (nm)	Thickness of TiO_2_ (nm)	Thickness of Cu (nm)	*R_min_* at *n_s_*		FWHM (deg) at *n_s_*		*δθ_SPR_* (deg) at *δn_s_* = 0.005	S_n_ (°/RIU)	FOM (RIU^−1^)
			1.33	1.335	1.33	1.335			
BaTiO_3_ (5 nm)	0	55	0.014	0.013	0.72	0.75	0.66	132	183
BaTiO_3_ (10 nm)	0	55	0.0097	0.0081	1.23	1.29	0.89	178	144.7
BaTiO_3_ (15 nm)	0	45	0.096	0.0095	4.27	4.65	2.43	486	113.8
BaTiO_3_ (5 nm)	5	55	0.0067	0.0061	0.69	0.71	0.66	132	191.30
BaTiO_3_ (10 nm)	5	55	0.0013	0.00053	1.15	1.21	0.90	180	156.5
BaTiO_3_ (15 nm)	5	45	0.026	0.031	4.03	4.49	2.76	552	136.97
Fe_2_O_3_ (5 nm)	5	55	0.0062	0.0072	0.96	1.003	0.71	142	147.9
Fe_2_O_3_ (10 nm)	5	50	0.017	0.034	3.28	3.6	1.41	282	85.97
Fe_2_O_3_ (11 nm)	5	45	0.0043	0.049	5.2	5.78	2.03	406	78.07
MoO_3_ (5 nm)	5	55	0.0046	0.0042	0.58	0.6	0.59	118	203.4
MoO_3_ (10 nm)	5	55	0.0014	0.001	0.75	0.77	0.65	130	173.3
MoO_3_ (27 nm)	5	45	0.013	0.0022	3.68	4.18	1.93	386	104.89

**Table 3 sensors-23-07495-t003:** The comparison of the proposed structure to the similar type of former reported SPR sensor structure.

Ref.	*λ* (nm)	Configuration	Sensitivity (°/RIU)	FOM (RIU^−1^)	DA (deg^−1^)
[60]	600	SF11-Cu-WSe_2_	92	-	-
[59]	633	CaF_2_-TiO_2_-Au-PtSe_2_	218	53.32	0.2466
[49]	633	BK7-TiO_2_-Au-Mxene-Antimonene	224.26	19.05	0.0849
[61]	633	BK7-Cu-Ni-BP-Ti_3_C_2_T_x_	304.47	57.85	0.19
[41]	633	FK51A-Ag-BaTiO_3_-BlueP/MoS_2_	347.82	60.52	0.174
[62]	632.8	BK7-Cr-Ag-BaTiO_3_-Go	372	88.11	-
[19]	633	CaF_2_-Cu-Fe_2_O_3_-Antimonene	398	-	-
[18]	633	BK7-Cu-Ni-BP-Antimonene	446	93.10	0.2
proposed	633	BK7-TiO_2_-Cu-MoO_3_	386	104.89	0.2717
proposed	633	BK7-TiO_2_-Cu-Fe_2_O_3_	406	78.07	0.1923
proposed	633	BK7-TiO_2_-Cu-BaTiO_3_	552	136.97	0.2481

## Data Availability

The data reported in this paper can be availed from the corresponding author upon request.

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
