# Peer review of "Numerical Investigation on High-Performance Cu-Based Surface Plasmon Resonance Sensor for Biosensing Application"

_sensors, 2023, doi:10.3390/s23177495_

Round 1
Reviewer 1 Report
Overall, this manuscript demonstrates a novel chip configuration (BK7-TiO2-Cu-BaTiO3) that shows high sensitivity of SPR. The thickness of the layers was optimized, resulting sensitivity as high as (552°/RIU). The developed SPR chips provide new perspectives for highly sensitive SPR as well as further biosensing applications. This is good work that is worth of publication in Sensors. However, the following concerns should be addressed before publication.
Major concerns:
1. Please annotate all the variables/constants in the equations. For example, “??r” and “???” in equation (1). For equation 10, please add corresponding labels for the fields H and E in the figure (such as figure 1).
2. In Figure 5 (a) and (b), with 5 nm TiO2 and 5/10 nm MoO3, the sensitivity reached a plateau in a certain range of Cu thickness (45~55). Can authors explain the observation?
Minor concerns:
1. Please keep using consistent name for the same concept. E.g., in abstract, please use TiO2-Cu-BaTiO3 instead of TiO2-copper-BaTiO3.
2. Please make sure that all the abbreviations are correct. Once a new concept is first introduced, please use full name as well as abbreviation. E.g., in abstract, FOM; in introduction, SPR.
3. Please carefully revise the figures. Make sure the labels are in the same size and font, also well aligned. E.g., Figures 2, 3, and 4.
Minor editing of English language required
Author Response
Thank you very much for your positive and constructive comments which allowed us to improve the quality of the manuscript.
Q1: Please annotate all the variables/constants in the equations. For example, “εmr” and “εmi” in equation (1). For equation 10, please add corresponding labels for the fields H and E in the figure (such as figure 1).
Reply: The above mentioned corrections are carried out in the revised manuscript.
Q2: In Figure 5(a) and (b), with 5nm TiO2 and 5/10nm MoO3, the sensitivity reached a plateau in a certain range of Cu thickness (45-55). Can authors explain the observation?
Reply: The plateau region is due to the small increment in sensitivity in these regions. The revised figure with scale range changed shows the incremental values precisely.
Q3: Please keep using consistent name for the same concept. E.g., in abstract, please use TiO2-Cu-BaTiO3 instead of TiO2-Copper-BaTiO3.
Reply: Corrections were carried out in accordance with the kind comment of the reviewer.
Q4: Please make sure that all the abbreviations are correct. Once a new concept is first introduced, please use full name as well as abbreviation. E.g., in abstract, FOM; in introduction, SPR.
Reply: The mentioned corrections are carried out in the revised manuscript.
Q5: Please carefully revise the figures. Make sure the labels are in the same size and font, also well aligned. E.g., Figure 2, 3, and 4.
Reply: The figures are revised as per the direction in the revised manuscript.
Reviewer 2 Report
This is an interesting manuscript but requires improvement before it can be published
Please see some of the minor concerns below:
- in the title please change SPR by Surface Plasmon Resonance
- please change the title by "numerical ..
- It is suggested that the abstract be improved to highlight the value of the proposed model, including rational out, problems, numerical values
-The reference cited in introduction section should be improved (1) A few relevant references need to be cited in this article to enrich the background like, . e.g., Structural, optical and sensing properties of Cr-doped TiO2 thin films. Sensor Letters, 2011, 9.5: 1697-1703.
- Comprehensive proofread is essential throughout the manuscript.
- justify the novelty of the proposed approach
- What about the surface rugosity effect of these devices
- In the conclusion, almost no perspectives are given for this work. Please develop
Author Response
Thank you very much for your positive and constructive comments which allowed us to improve the quality of the manuscript.
Q1: in the title please change SPR by Surface Plasmon Resonance.
Reply: According to the kind comment of the reviewer, the corrections are carried out in the revised manuscript.
Q2: please change the title by “numerical.
Reply: The above mentioned correction was carried out in the revised manuscript.
Q3: It is suggested that the abstract be improved to highlight the value of the proposed model, including rational out, problems, numerical values.
Reply: The abstract is modified as per the kind direction.
Q4: The reference cited in introduction section should be improved (1) A few relevant references need to be cited in this article to enrich the background like, .e.g., Structural, optical and sensing properties of Cr-doped TiO2 thin films. Sensor Letters, 2011, 9.5: 1697-1703.
Reply: The above reference is included in the revised manuscript.
Q5: Comprehensive proofread is essential throughout the manuscript.
Reply: The revised manuscript was carefully proofreaded.
Q6: Justify the novelty of the proposed approach
Reply: The statement exhibiting the novelty of the proposed work is included in the revised manuscript.
Q7: What about the surface rugosity effect of these devices
Reply: Justification statements are included in the revised manuscript.
Q8: In the conclusion, almost no perspectives are given for this work. Please develop.
Reply: As per the kind direction of the reviewer, the conclusions were modified in the revised manuscript.
Reviewer 3 Report
- In this paper, authors introduced "numerical analysis" of high performance CU based SPR sensor by biosensing application. The numerical simulation i s meaningful. However, following issues should be addressed.
1. Title should be changed to "numerical approach" of Cu based SPR... This paper did not deal with any experiment.
2. There is a very limited portion of biosensing part description using fourth layer. Out of BaTiO3, Fe2O3, and MoO3, which one has advantages with biosample? This kind of content was missing, which is critical based on author's tiltle.
3. In introduction part of line 43~45, the description about gold (Au) was wrong. Adhesion or Au-Sulfur bond of biomolecules on Au layer is extremely advantageous in biosensor.
4. Figure 4 and Figure 5 can be placed as supplementary information with Fe2O3 and MoO3 layers.
5. In results and discussions at lines of 153~161, there should be proper reference papers.
6. Even though the paper is dealing with simulation, the method of BaTiO3, Fe2O3, and MoO3 film formation on Cu should be discussed with appropriate references.
N/A
Author Response
Thank you very much for your positive and constructive comments which allowed us to improve the quality of the manuscript.
Q1: Title should be changed to “numerical approach” of Cu based SPR. This paper did not deal with any experiment.
Reply: The title has been corrected as requested by the reviewer.
Q2: There is a very limited portion of biosensing part description using fourth layer. Out of BaTiO3, Fe2O3 and MoO3, which one has advantageous with biosample? This kind of content was missing, which is critical based on author’s title.
Reply: The biosensing part is elaborated in the revised manuscript.
Q3: In introduction part of line 43-45, the description about gold (Au) was wrong. Adhesion or Au-Sulfur bond of biomolecules on Au layer is extremely advantageous in biosensor.
Reply: The necessary reference is included in the revised manuscript
Q4: Figure 4 and Figure 5 can be placed as supplementary information with Fe2O3 and MoO3 layers.
Reply: We expect those figures are needed for simultaneous comparisons.
Q5: In results and discussions at lines of 153-161, there should be proper reference papers.
Reply: As per the direction of the reviewer, the references are included in the results and discussions part.
Q6: Even though the paper is dealing with simulation, the method of BaTiO3, Fe2O3, and MoO3 film formation on Cu should be discussed with appropriate references.
Reply: According to the request of the reviewer, inclusion of the method of BaTiO3, Fe2O3, and MoO3 film formation on Cu with appropriate references in the revised manuscript was added.
Round 2
Reviewer 1 Report
The revised the manuscript has already addressed the concerns in the previous review comments. The description on the data matches the experimental design. In addition, more details have been added in the results section, which provides more comprehensive discussion. Overall, the revised manuscript is suitable for publication in Sensors.
Reviewer 3 Report
- Most of issues raised by reviewer were resolved.
- N/A